# A Review on Epidemiological and Clinical Studies on Buckwheat Allergy

**DOI:** 10.3390/plants10030607

**Published:** 2021-03-23

**Authors:** Dan Norbäck, Gunilla Wieslander

**Affiliations:** Department of Medical Science, Uppsala University, SE 75185 Uppsala, Sweden; gunilla.wieslander@medsci.uu.se

**Keywords:** buckwheat, buckwheat allergy, occupational, asthma, diet, buckwheat husk pillow, anaphylaxis

## Abstract

Background: Cultivated buckwheat include two species originating from China: common buckwheat (*Fagopyrum esculentum*) and tartary buckwheat (*Fagopyrum tartaricum*). Buckwheat can cause IgE-mediated allergy, including severe allergic reactions and anaphylaxis. Exposure can occure when eating buckwheat food (food allergen), when producing or handling buckwheat food (occupational exposure) or when sleeping on buckwheat husk pillows (houeshold environmental exposure). Methods: A search on buckwheat allergy in the medical datbase PubMed from 1970–2020. Result: A number of allergenic proteins have been identified in common buckwheat (e.g., Fag e 1, Fag e 2 and Fag e 3) and in tartary buckwheat (e.g., Fag t 1, Fag t 2, Fag t 3). Clinically relevant cross-reactivity has been described between buckwheat and peanut, latex, coconut, quinoa, and poppy seed. The prevalence of buckwheat allergy in the population can be estimated as 0.1–0.4% in Japan, Korea and buckwheat consuming areas of China. Among patients in allergy clinics in different countries, 2–7% has confirmed buckwheat allergy. School studies from Japan and Korea found 4–60 cases of buckwheat-related anaphylaxis per 100,000 school children. The incidence of severe allergic reactions to buckwheat, including anaphylaxis, can be estimated as 0.1–0.01 cases per 100,000 person-years. Conclusions: Buckwheat allergy is a neglected allegy deserving further attention but severe allergic reactions are rare.

## 1. Introduction

There are two species of cultivated buckwheat, common buckwheat (*Fagopyrum esculentum*) and tartary buckwheat (*Fagopyrum tartaricum*). Buckwheat is used in different types fo food such as noodles, bread, pancakes, porridge and kasha, pre-boiled and dried whole buckwheat grain. Common buckwheat is dominating globally, but in some areas, e.g., in Shanxi province in China, both types of buckwheat are cultivated and consumed [1]. In recent years, tartary buckwheat tea has become popular, especially in China, because of its high antioxidant content linked to the polyphenol compounds rutin and quercetin [2]. Buckwheat allergy is an IgE-mediated allergy, sometimes causing severe allergic reactions [3,4]. Allergic reactions can occur when eating buckwheat food products, in the work environment or when sleeping on pillows containing buckwheat husks. The first review article on buckwheat allergy was published in 1996 [3]. More recent review articles on buckwheat allergy have focused on buckwheat allergy in Europe [5], in Australia [6] or in humans as well as in domestic animals [7]. The aim of this publication is given an updated overview of different aspects of buckwheat allergy.

## 2. Materials and Methods

Scientific publications on buckwheat allergy published in the past 50 years (1971–2020) were searched in the medical database PubMed. The search term was “buckwheat allergy”. Moreover, some key publications published before 1971 were included, obtained from the reference list of the first review article on buckwheat allergy [3]. Most publications on buckwheat allergy were published after 2000, indicating an increased concern about this plant allergy. In total, 166 articles were identified by the PubMed search, including nine review articles, six clinical trials and one editorial. Publications mentioning “buckwheat allergy” but did not include a study buckwheat allergy in patients or in population samples were excluded, except for a few important review articles. Food allergy (117 articles), anaphylaxis (46 articles) and asthma (41 articles) were the most common medical disorders mentioned in the abstract, title or keywords. Dermatitis (14 articles), rhinitis (11 articles), occupational buckwheat allergy (17 articles), occupational asthma (12 articles) and buckwheat pillow allergy (9 articles) were less often mentioned in the articles (Table 1).

## 3. Results

### 3.1. Early Studies on Buckwheat Allergy

The first case of buckwheat allergy was described in 1909. It was an adult with asthmatic symptoms, rhinitis, urticaria and angioderma provoked by eating small amounts of buckwheat flour [8]. In 1926, skin prick testing with an extract of buckwhat was used for the first time to diagnose buckwheat allergic asthma in a sensitized child [9]. Later, it was found that 27 patients referred to allergy clinics in the USA had a positive skin prick test to buckwheat [10].

### 3.2. Identification of Buckwheat Allergens

In an early study from 1994, a 24 kilo Dalton (kDa) major allergenic protein (BW24KD) was identified in common buckwheat seeds [11]. Later, a study from Korea verified that this protein is a major allergen in common buckwheat and in addition other smaller allergenic proteins (9, 16 and 19 kDa) were identified [12]. Moreover, three larger proteins (40, 45 and 48-kDa) were found to be important buckwheat allergens [13]. Nowadays, allergens in common buckwheat are reported as Fag e 1 (13S globulin), Fag e 2 (2S globulin), Fag e 3 (7S globulin/vicilin), Fag e 4 (antimicrobial peptide), Fag e 5 (vicilin-like protein), Fag e 10 kD (α-amylase inhibitor/tripsin inhibitor) and Fag e T1 (trypsin inhibitor) [5,7]. Fage 1 and Fag e 2 are considered to be the main allergens in common buckwheat [14]. Moreover, Fag e 3, Fag e 4 and Fag e 5 are relevant allergens in buckwheat allergy [15]. Fag e 2 seems to be important for the cross-reactivity between buckwheat and latex and concomitant sensitization to legumin [15].

Some work has been done to identify allergenic proteins in tartary buckwheat [16]. There are three main allergenic proteins identified, Fag t 1, Fag t 2 and Fag t 3. Fag t 1 is a legume-type protein, and a major allergen in tartary buchwheat [17]. Moreover, Fag t 2 (a 16 kDa protein), has been identified in tartary buckwheat seeds. Fag t 2 belongs to the 2S albumin family and is expressed in the embryo of tartary buckwheat seeds [18]. Another major allergen in tartary buckwheat is Fag t 3, a seed storage protein belonging to the cupin superfamily [19].

### 3.3. Diagnosis of Buckwheat Allergy

Buckwheat allergy can be diagnosed either by skin prick testing, by measuring specific immunoglobulin E (IgE) or by oral food challenge (OFC). OFC is the golden standard to diagnosing food allergies. One review concluded that specific IgE against Fag e 3 was reported to be a good marker of buckwheat food allergy [20]. An allergy test measuring specific IgE antibodies to Fag e 3 has been developed in Japan [21]. In recent years OFC has been used to diagnose buckwheat allergy. Among 72 patients in Italy with suspected buckwheat allergy, 30 (41.7%) were sensitized to buckwheat and 24 (33%) had a positive OFC test with buckwheat flour [22]. In another OFC study on 126 patients in Japan, only 18 (14%) had positive OFC results to buckwheat [23]. Since OFC but can cause anaphylaxis, it is important to predict the risk of anaphylaxis after OFC challenge to buckwheat. In an OFC buckwheat study on 60 patients in Japan, Fag e 3 specific serum IgE was the only tested factor predicting positive OFC results, and moreover, it could predict OFC-induced anaphylaxis. Among the 60 patients, 20 (33%) had positive OFC results and seven (12%) got anaphylaxia [24]. Another Japanese study found that a positive skin prick test with buckwheat extract was more useful to predict a positive OFC than specific serum IgE against buckwheat measured by the ImmunoCap assay system [25].

Food protein-induced enterocolitis syndrome (FPIES) is a no-IgE-mediated gastrointestinal food allergy causing repeated vomiting, diarrhea and abdominal pain. One case of FPIES caused by buckwheat has been reported, a four-year old boy in Japan. The diagnosis was verified by oral food challenge (OFC) and a lymphocyte stimulation test (LST). Skin prick test and specific IgE to buckwheat were all negative [26]. We found no other studies on non-IgE-mediated buckwheat allergy.

### 3.4. Cross-Reactivity between Buckwhat and Other Plants

There is some cross-reactivity between buckwheat and other food allergens. Clinically relevant cross-reactivity have been found for latex [27,28], coconut [29], quinoa [30], peanut [31], and poppy seed [32,33]. We found no clinical studies on cross-reactivity between common and tartary buckwheat, but such cross-reactivity between these two species of buckwheat could be expected.

### 3.5. Buckwheat Allergy and Fagopyrism in Animals

Buckwheat products are sometimes used to feed horses, so-called “horse muesli” [7]. In one study on specific IgE activity in sera from 51 horses from Europe and Japan, the most prevalent allergy in horses with allergic reactions was against buckwheat. A total of 72.5% of the horses (37/51) has specific IgE against the buckwheat allergen Fag e 2 [34]. Moreover, a murin experimental model for buckwheat allergy has been developed to study immunomodulatory effects of phosphorylated hypoallergenic Fag e 2. It was found that this attenuated allergen can suppress Th2-induced allergic responses [14]. Fagopyrism is a rare phototoxic reaction described in animals eating buckwheat plants. Fagopyrism is a toxic reaction, not an allergy. It can come after exposure to sunlight of albino or partially pigmented animals such as swine, cattle, horses, sheep or rabbits. Fagopyrism was first described by Bruce in 1917 [35]. He noted phototoxic reactions and intoxicated behavior in Yorkshire (white) pigs that had grazed on common buckwheat in bloom. The chemical structures of the fagopyrins have remained unclear for a long time but recently analytical methods for buckwheat fagopyrins have been developed. One research group identified at least six different fagopyrins in buckwheat and identified two protofagopyrin that can be transformed into fagopyrins at light exposure [36].

### 3.6. Occupational Buckwheat Allergy

Occupational buckwheat asthma when making buckwheat noodles or pancakes/creps has been described in early case-reports from Japan [37], Korea [38] and different countries in Europe [39,40,41]. In a recent publication, six cases of occupational immediate allergy to buckwheat were described. They were three cooks, two bakers, and one worker in a grocery store exposed to buckwheat flour at work. Four were diagnosed with occupational asthma, four with occupational rhinitis and two with occupational contact urticaria caused by buckwheat. Three of the patients with occupational buckwheat allergy got anaphylactic reactions after eating buckwheat food [42]. A similar allergy pattern was observed in an early case report. One woman first sensitized to buckwheat when manufacturing buckwheat husk pillows got an anaphylactic reaction when eating buckwheat crepes four years later [43]. However, few epidemiological studies exist on occupational buckwheat allergy. One early study from Sweden found a high prevalence (28%) of clinically diagnosed buckwheat allergy among workers inhaling dust from buckwheat when repacking buckwheat grains and buckwheat flour. The allergic reactions included asthma, rhinitis and dermatitis. Atopic heredity was uncommon. Total airborne dust exposure was 1.7 mg/m^3^ when repacking buckwheat, lower than existing permissible exposure limit values (PEL) in Sweden for organic dust (5 mg/m^3^) [44]. Another study from China investigated buckwheat allergy among 69 subjects eating buckwheat food or working with buckwheat products in a noodle factory in the city of Taiyuan. Taiuyuan is the capital of the Shanxi province in China, an area growing and consuming buckwheat. One male industrial worker (1.4%) had a positive skin prick test to buckwheat but reported no symptoms when eating or handing buckwheat products [1]. Moreover, one Japanese study investigated the efficiency of using dust respirators at work in a noodle factory. In this study, one of the workers had developed occupational asthma from buckwheat. The patient could return to work without respiratory problems when wearing a dust respirator [45].

### 3.7. Domestic Exposure to Buckwheat Allergens from Buckwheat Husk Pillows

Pillows filled with buckwheat husk have been popular in China and Korea for a long time but are nowadays used in other parts of the world as well. These pillows can cause domestic exposure to airborne buckwheat allergens when sleeping on the pillows. One Korean study described three cases of childhood nocturnal asthma triggered by buckwheat allergens from buckwheat husk pillows. All three children had a positive skin prick test to buckwheat and none had a house dust mite allergy. The nocturnal asthmatic symptoms were improved after removing the pillows [46]. One study from USA reported a case of a person developing asthma and worsening of existing allergic rhinitis after exposure to buckwheat husk pillows. The patient was diagnosed with buckwheat allergy and house dust mite allergy. House dust mite avoidance measures did not improve the symptoms but after removal of the buckwheat husk pillows, asthma symptoms disappeared and the rhinitis symptoms were improved [47]. One study from an allergy clinic in Beijing, China, identified seven patients with buckwheat allergies. Five of the patients used buckwheat husk pillows and the authors concluded that such pillows can be an important route of exposure to buckwheat allergens in China [48]. Finally, one study compared the concentration of endotoxin and house dust mite allergen levels (Der f 1) in dust from synthetic pillows and buckwheat husk pillows. Levels of Der f 1 were low and similar in both types of pillows but buckwheat husk pillows contained 12 times higher levels of endotoxin as compared to synthetic pillows. The authors concluded that buckwheat pillows can cause high endotoxin exposure that could influence asthma severity of atopic asthmatics [49].

### 3.8. Hospital-Based Studies on Buckwheat Allergy

Studies on consecutive cases of buckwheat allergy in patients referred to allergy clinics have been published. These studies can be used to estimate the proportion of all allergy clinic patients that have buckwheat allergies. One early study from 1931 found that 5.4% of all consecutive patients in an allergy clinic in the USA had verified buckwheat allergy [10]. In the 1960′ies, a nationwide survey of buckwheat allergy in allergy clinic patients found 169 cases of buckwheat allergy in Japan. Most (86%) were young children with food allergies or respiratory buckwheat allergy. The most common allergic reaction to buckwheat was asthma (82% of all cases) and 18 cases (11%) had an anaphylactic reaction [50]. In a later study from Beijing, seven out of a total of 192 patients diagnosed with food allergy (3.6%), were sensitized to buckwheat [48]. In a study from Korea, including 415 adult patients with a diagnosed food allergy, 31 (7.4%) had buckwheat allergy [51]. In another patient study from Korea, immediate-type food allergy was investigated in 1353 children (0–18 y). Most children (65%) were less than 2 years old. In total, 1.9% of all children had buckwheat allergy; 0.9% among those <2 year, 3.0% among those 2–6 year, 13.2% among those 7–12 year and 17.2% among those 13–18 year [52]. A nationwide survey of food allergies in Japan estimated that buckwheat allergy accounted for 2.2% of all immediate-type food allergies [53]. In a nationwide study on 1954 allergy patients in Italy, 70 patients (3.6%) were diagnosed with buckwheat allergy [54].

### 3.9. Epidemiological Population Studies on Buckwheat Allergy

Some large population studies are available on the prevalence of buckwheat allergy in the general population. The Yokohama study included 92,680 schoolchildren and estimated that the prevalence of reported buckwheat allergy was 0.2% among schoolchildren in Yokohama [55]. Another study among 10,192 school children and university students (13–21 year) in Taiyuan, China, found that 0.4% reported buckwheat allergy [56]. In Sweden, subjects with coeliac disease eat buckwheat products since buckwheat does not contain gluten. Among 870 members of a society for subjects with coeliac disease in the Stockholm area, 3.9% reported buckwheat allergy [56]. A school study from Korea included 27,435 elementary school children (6–12 year) and 14,777 middle school children (12–15 year). They found that 25 elementary school children (0.09%) and 19 middle school children (0.13%) reported buckwheat allergy [57]. In a later study among 29,843 school children in Korea, performed in 2015, 0.13% of all children reported buckwheat allergy. If stratified by age, 0.03% among those 6–7 year, 0.10% among those 9–10 year, 0.17% among those 12–13 year and 0.18% among those 15–16 year reported buckwheat allergy [58].

### 3.10. Case Reports on Anaphylaxis Triggered by Buckwheat

Buckwheat allergy can cause severe asthmatic reactions and anaphylaxis. A number of report cases on anaphylaxis triggered by buckwheat have been reported. One early study described a patient who developed anaphylaxis, with urticarial and hypertension, after eating buckwheat crepes. [43]. In the USA, a patient had two episodes of anaphylaxis after eating buckwheat. He was highly allergic to buckwheat and had previously experienced allergic asthma, rhinitis, conjunctivitis and contact urticaria after sleeping on a buckwheat pillow [59]. In United Kingdom (UK), one patient got anaphylaxis after eating home-baked bread containing buckwheat flour [60]. Three case reports from Germany described buckwheat-induced anaphylaxis. One nurse with known latex allergy and buckwheat allergy got anaphylaxis after eating a muesli bar containing buckwheat [61]. Another patient with buckwheat allergy got an anaphylactic reaction after eating food containing poppy seeds. She had cross-reactivity between buckwheat and poppy seed [32]. A third patient with buckwheat allergy got a life-threatening anaphylactic reaction after eating galettes, a special French buckwheat pancake [62]. In Austria, a 7 year old boy got anaphylaxis after eating a cake containing buckwheat flour. He had previously got severe allergic reactions to hazelnuts and suffered from an oral allergy to poppy seeds. Buckwheat allergy was confirmed but also cross-reactivity between an 11S globulin in buckwheat and hazelnut and poppy seed [33]. In Italy, one patient got anaphylaxis on four different occasions after eating pizza containing hidden buckwheat flour. Allergy testing to buckwheat was negative but a double-blind placebo-controlled food challenge test with buckwheat flour was positive [63]. In Taiwan, one patient developed anaphylaxis after eating buckwheat porridge. She denied any previous ingestion of buckwheat or any exposure to buckwheat pillows but had confirmed buckwheat allergy [64]. Finally, two early publications reported on two cases of fatal buckwheat anaphylaxis. One fatal case of buckwheat allergy occurred after eating “nyan-mien”, a Korean buckwheat noodle [65]. Moreover, an 8 year old girl in Japan developed anaphylaxis after eating buckwheat noodles (“zaru soba”). After eating the noodles for lunch she had sports class and swam 50 m in a swimming pool as fast as she could and then developed exercise-induced buckwheat anaphylaxis within 30 min. Her buckwheat allergy was confirmed after death [66].

Since buckwheat allergy is an immediate type of allergy, anaphylaxis usually comes quickly after exposure (within one hour). However, one case-report from Taiwan describes biphasic buckwheat anaphylaxis, with a second delayed reaction coming within a day after being exposed to buckwheat [67]. This suggests that the anaphylactic patient should be monitored for a longer period, at least 6–8 h and if having hypotension for at least 12–24 h [67].

### 3.11. Anaphylaxis in Patient Studies

There are some studies available on the prevalence of buckwheat-related anaphylaxis in consecutive cases of allergy patients. In one study among 1110 food-allergic patients in 19 allergy centers in Italy, only one patient (0.1%) had an anaphylactic reaction to buckwheat [68]. In a study from Korea in 415 adult food allergy patients, eight patients (1.9%) had anaphylaxia triggered by buckwheat [51]. In another hospital-based study from Korea on 740 children with food-related anaphylaxis, 48 (6.5%) of all anaphylactic reactions were triggered by buckwheat allergy [69].

### 3.12. Epidemiological Studies on Severe Allergic Reactions and Anaphylaxis from Buckwheat

There are few population-based studies on the prevalence or incidence of buckwheat-induced anaphylaxis. One early study among 92,680 school children in Yokohama, Japan, found that four children (0.004% or 4 prevalent cases per 100,000 children) had anaphylaxis needing medical emergency treatment after eating buckwheat [55]. In a prevalence study in school children in Korea, 0.06% or 60 prevalent cases per 100,000 children, had experienced buckwheat-induced anaphylaxis [58]. In a study from Finland, the incidence of severe allergic reactions in the general population in the city of Helsinki was estimated. During an eight-year follow-up period (2000–2007), they identified 530 severe allergic reactions in the population, equivalent to an annual incidence of 1 case per 100,000 person-year (0.001%). The reaction was life-threatening anaphylaxis in 26% of the cases with no death reported. Five of these 530 severe allergic reactions (1%) were due to buckwheat allergy [70]. From these data, it can be estimated that the incidence of severe allergic reactions to buckwheat in Helsinki can be around 0.01 cases per 100,000 person-year. A study from Korea based on the Korean Health Insurance Review (KHIRA) found that the total incidence of anaphylaxis among Korean children (<19 year) was 0.7–1 cases per 100,000-person-years. Food was the most common cause of anaphylaxis (25.9%). No deaths were reported. They reported that buckwheat was among the most common causes of anaphylaxis [71].

### 3.13. Buckwheat Food Allergy in Relation to Allergy to Other Cereals or Pseudocereals

Most consecutive patient studies or population studies on buckwheat food allergy cited in this review do not mention other types of food allergies. However, four studies from Korea [51,52,57,58] and one from Japan [53] report on food allergy to other types of cereals or pseudocereals. One population study from Korea, including 27,435 elementary school children (6–12 year) and 14,777 middle school children (12–15 year), found that in year 2000, 0.09% of elementary school children reported buckwheat allergy while only 0.04% reported wheat allergy. In middle school children, 0.13% reported buckwheat allergy and 0.08% wheat allergy [57]. In a later population study among 29,843 school children in Korea, performed in 2015, 0.13% reported buckwheat allergy while only 0.04% reported wheat allergy [58]. In a patient study among children in Korea, 1.9% of all food allergy patients had buckwheat allergy, 7.5% had wheat allergy and 0.2% barley allergy [52]. In another study among adult food allergy patients in Korea, 7.4% had buckwheat allergy 15.1% had a wheat allergy and 1.1% rice allergy [51]. Finally, in the nationwide study in Japan on patients with an immediate-type food allergy, buckwheat allergy was 2.2% and wheat allergy was 11.7% [53].

## 4. Discussion

Buckwheat can cause IgE-mediated allergy and in rare cases severe allergic reactions such as asthma attacks and anaphylaxis. In rare cases, a biphasic allergic reaction can occur. Buckwheat allergy can be a food allergy, occupational allergy or a domestic allergy to buckwheat husk pillows. Most of the 166 scientific publications identified in this review focused on food allergy, anaphylaxis and asthma, fewer publications mention occupational buckwheat allergy or allergy to buckwheat husk pillows. A number of allergenic proteins have been identified in common buckwheat (e.g., Fag e 1, Fag e 2 and Fag e 3) and in tartary buckwheat (e.g., Fag t 1, Fag t 2, Fag t 3). Clinically relevant cross-reactivity has been described between buckwheat and latex, coconut, quinoa, peanut, and poppy seed. Hospital-based studies from different countries have found that 2–7% of all patients in allergy clinics have confirmed buckwheat allergy. Thus, buckwheat allergy is not a main food allergen but can be relatively common among allergy patients in some countries. This suggests that allergy doctors shall be aware of buckwheat allergy. The prevalence of buckwheat allergy in the population can be estimated to be 0.1–0.4% in Japan, Korea and buckwheat consuming areas of China but higher in certain subgroups such as subjects with coeliac disease. In the population in Korea and Japan, reports on buckwheat allergy can be more common than reports on wheat allergy. However, among patients referred to allergy clinics in Korea and Japan, buckwheat allergy can be less common than wheat allergy but more common than allergy to rice and barley. Anaphylaxis is a severe allergic reaction and buckwheat-triggered anaphylaxis has been reported in a number of case studies, including two fatal cases. One early Japanese study found a prevalence of anaphylaxis among school children of 4 cases per 100,000 persons. A later study from Korea found that the prevalence of anaphylaxis in school children was 60 cases per 100,000 children The incidence of severe allergic reactions to buckwheat, including anaphylaxis, can be estimated to be 0.1–0.01 cases per 100,000 person-years.

## Figures and Tables

**Table 1 plants-10-00607-t001:** The total number of articles (N = 166), and types of articles identified in PubMed from 1970–2020 by the search term “buckwheat allergy”.

Type of Scientific Article	
Journal article	155
Review article	9
Clinical trial	6
Editorial	1
Meta-analysis	0
Total number of articles	166
**Type of Medical Disorders Mentioned in the Article ***	
Food allergy	117
Anaphylaxis	46
Asthma	41
Respiratory disease	29
Occupational allergy	17
Dermatitis	14
Occupational asthma	12
Rhinitis	11
Buckwheat pillow allergy	9
Occupational rhinitis	6
Occupational dermatitis	2
Fatigue	2

* One article can describe more than one medical disorder and can be of more than one type.

## Data Availability

Not applicable (a review).

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
