# Peer review of "A Review on Epidemiological and Clinical Studies on Buckwheat Allergy"

_plants, 2021, doi:10.3390/plants10030607_

Round 1

Reviewer 1 Report

The paper reviews all information about buckwheat allergy and gives an update on buckwheat allergy in humans and animals.

As the paper is based on a literature survey in PubMed, it would be interesting to mention the number of papers identified and used in this work and maybe mention the number of papers per theme or type of allergy investigated in this review (could be presented as a figure or a table)... This could give information about the most analyzed type of buckwheat allergy and where the research efforts are concentrated.

In the discussion, I would suggest discussing buckwheat allergy regarding other types of food allergy to give an idea of the importance of buckwheat allergy compared to other food and grain allergy.

Minor comments are included in the manuscript

Author Response

Reviewer 1

The paper reviews all information about buckwheat allergy and gives an update on buckwheat allergy in humans and animals. As the paper is based on a literature survey in PubMed, it would be interesting to mention the number of papers identified and used in this work and maybe mention the number of papers per theme or type of allergy investigated in this review (could be presented as a figure or a table)... This could give information about the most analysed type of buckwheat allergy and where the research efforts are concentrated.

Response to comment:

We have included a table with total number of buckwheat allergy articles found in PubMed from 1970-2020 (N=166), and types of medical disorders mentioned in the articles.

In the discussion, I would suggest discussing buckwheat allergy regarding other types of food allergy to give an idea of the importance of buckwheat allergy compared to other food and grain allergy.

Response to comment:

We have added a paragraph in results, adding results on other cereal/pseudocereal food allergy if such data has been reported in the consecutive patient studies or the population based epidemiological studies cited in this review. This issue is also mentioned in the discussion.

Minor comments are included in the manuscript

Response to comment: We have corrected these minor error in the text. Moreover we have added information on type of protein for Fag t 2 and Fag 3.

Reviewer 2 Report

Dear authors,

The authors in the paper entitled: "A review on epidemiological and
clinical studies on buckwheat allergy” described differ buckwheat
products, which nowdays are produce arround the world. Buckwheat is
common known as a plant with high pro-healthy activity and is
recommended for food production and consumption by Nutritionists.
However, there is still lack of evidences about the harm activity of
food buckwheat products. The authors introduced very interesting
epidemiological information aboutcases of buckwheat sensitized.
Furtheremore they discussed different way of buckwheat alergies, such
as: foodmallergy and inhalation.
They described sympthoms also, such as: rhinitis, urticaria, and also
anaphylactic reactions. Epidemiological cases were intoduced due to
previously papers with good impact for develop of science. In my opinion
the paper was written in good style, and in the future will be usefull
for scientist interested in the scientific area of buckwheat.

i am pleased to inform you that the paper is very well described, however i found some minor lapses. See attached pdf file, you can find there minor comments.

Author Response

Reviewer 2

The authors in the paper entitled: "A review on epidemiological and
clinical studies on buckwheat allergy” described differ buckwheat
products, which nowadays are produce around the world. Buckwheat is
common known as a plant with high pro-healthy activity and is
recommended for food production and consumption by Nutritionists.
However, there is still lack of evidences about the harm activity of
food buckwheat products. The authors introduced very interesting
epidemiological information about cases of buckwheat sensitized.
Furthermore they discussed different way of buckwheat allergies, such
as: food allergy and inhalation.
They described symptoms also, such as: rhinitis, urticaria, and also
anaphylactic reactions. Epidemiological cases were introduced due to
previously papers with good impact for develop of science. In my opinion
the paper was written in good style, and in the future will be useful
for scientist interested in the scientific area of buckwheat.

I am pleased to inform you that the paper is very well described, however I found some minor lapses. See attached pdf file, you can find there minor comments.

Response to comment: We have corrected these minor error in the text